# Optimal Encoding in Stochastic Latent-Variable Models

**DOI:** 10.3390/e22070714

**Published:** 2020-06-28

**Authors:** Michael E. Rule, Martino Sorbaro, Matthias H. Hennig

**Affiliations:** 1Department of Engineering, University of Cambridge, Cambridge CB2 1PZ, UK; mer49@cam.ac.uk; 2Institute of Neuroinformatics, University of Zürich and ETH, 8057 Zürich, Switzerland; martino@ini.uzh.ch; 3Institute for Adaptive and Neural Computation, School of Informatics, University of Edinburgh, Edinburgh EH8 9AB, UK

**Keywords:** information theory, encoding, neural networks, sensory systems

## Abstract

In this work we explore encoding strategies learned by statistical models of sensory coding in noisy spiking networks. Early stages of sensory communication in neural systems can be viewed as encoding channels in the information-theoretic sense. However, neural populations face constraints not commonly considered in communications theory. Using restricted Boltzmann machines as a model of sensory encoding, we find that networks with sufficient capacity learn to balance precision and noise-robustness in order to adaptively communicate stimuli with varying information content. Mirroring variability suppression observed in sensory systems, informative stimuli are encoded with high precision, at the cost of more variable responses to frequent, hence less informative stimuli. Curiously, we also find that statistical criticality in the neural population code emerges at model sizes where the input statistics are well captured. These phenomena have well-defined thermodynamic interpretations, and we discuss their connection to prevailing theories of coding and statistical criticality in neural populations.

## 1. Introduction

The rate at which information can be conveyed by a finite neural population is limited. Neurons have a maximum firing rate, and spiking communication is affected by noise. To utilize sensory information, the brain must find efficient coding strategies [1]. The spiking output of a neural population can therefore be viewed as a noisy communication channel. How might such a channel structure its available ‘code words’ to communicate diverse stimuli reliably? In conventional communications channels, deriving optimal codes is straightforward: the channel bandwidth is equal to the nominal bandwidth minus the entropy of any noise on the channel. The optimal code-word allocation is given by entropy coding [2], in which the cost (in bits) of a symbol with probability *p* should be roughly $-\log_{2}\left(p\right)$. Optimal coding strategies are more subtle in spiking channels, since the amount of noise depends on the symbol being transmitted: spiking variability is higher when neurons spend more time close to firing threshold. In addition, limited encoding bandwidth favours models that capture salient latent causes underlying sensory inputs [3,4,5].

Probabilistic sensory encoding is an area of active research. Broadly, incoming stimuli are known to suppress neuronal variability [6]. Some theories also suggest that variability reflects a sampling-based approach to representing statistical uncertainty [7]. However, variability need not imply uncertainty: the brain might also employ robust coding strategies, in which several noisy states are representationally equivalent [8,9]. Likewise, as we will show here, variability might be high not because the brain is uncertain, but because relatively less information is required to encode certain stimuli.

In this work, we used Restricted Boltzmann Machines (RBMs) to study encoding in stochastic spiking channels. RBMs consist of two layers of interconnected binary units, and the activity of these units is likened to the presence (in the case of a ‘1’) or absence (in the case of a ‘0’) of a spiking event in a single neuron in a specified brief time window. This simplified view of neurons as stochastic variables with discrete states also underlies ubiquitous mean-field models of neural population dynamics [10].

RBMs balance biological realism and theoretical accessibility. The stochastic and binary nature of RBMs resembles physiological constraints on spiking communication, while the interpretation of RBMs as Ising spin models also allows access to information-theoretic and thermodynamic quantities [11,12,13,14]. Although RBMs can be made to resemble spiking systems by extending their dynamics in time [15], statistical models of spiking populations commonly consider the synchronous case [16], which models only zero-lag dependencies between spikes. RBMs have been used as a statistical model to study retinal population coding [17,18], and multi-layer RBMs have been used as a model of retinal computation [19]. RBMs can be connected to more biologically plausible spiking models [20], but added biological plausibility does not not change essential statistical features that we wish to study, and obscures relevant thermodynamic interpretations.

The analysis of these models from a thermodynamic viewpoint leads to the observation of scale-free (‘1/f’) statistics in the frequencies of the population codewords evoked by incoming stimuli. These scale-free statistics are often interpreted as a correlate of a phenomenon called ‘statistical criticality’, whose significance is a topic of debate in recent literature. It is important to qualify the sense in which statistical criticality is explored in this manuscript. Critical statistics emerge naturally in large latent-variable models that are trained to represent external signals [21,22]. They are not scientifically meaningful in isolation [23], and can arise from many other causes [24,25]. In this work, the relevant connection is the absence of critical statistics in models that are too small to encode their inputs, and the emergence of these statistics above a certain model size.

We organize this work as follows. We first detail an RBM model of sensory encoding and present evidence of an optimal population size for capturing stimulus statistics. We then show that stimulus-dependent suppression of ‘neuronal’ variability is an essential feature of the learned encoding strategy. We observe that successful encoding corresponds to statistical criticality in the population code, a feature that is not inherited from the stimulus statistics. By examining a thermodynamics interpretation of the RBM, we show that statistical criticality connects to the optimization of the underlying network parameters, and that it suggests an optimal model size that balances accuracy verses the number of neurons used for encoding. We conclude with a discussion of the connection between the statistical machine-learning approach used here and other prevailing theories of sensory encoding.

## 2. Results

### 2.1. RBMs as a Statistical Machine-Learning Analogue of Stochastic Spiking Communication

Restricted Boltzmann Machines (RBMs; Figure 1a) are stochastic binary neural networks used in statistical machine learning [26]. They consist of two populations of stochastic binary units. One population, the ‘visible’ layer, is driven by incoming sensory stimuli. The other population, a ‘hidden’ layer, learns to encode the latent causes of these stimuli. These hidden units can therefore be interpreted as a stochastic spiking communication channel that conveys information about incoming stimuli.

Specifically, RBMs are a plausible abstract model of sensory encoding under the following assumptions: (1) Neurons have spatially localised (in stimulus space) receptive fields, and transmit information downstream using patterns of spiking. (2) Sensory processing extracts the latent causes underlying stochastic inputs. (3) Neuronal output can be considered over time-bins of finite width Δt, and population spiking output can therefore be viewed as consisting of binary codewords. (4) These binary outputs are stochastic. (5) Stimuli are communicated synchronously: every stimulus must be encoded within Δt amount of time, and must be communicated over a fixed number of output cells.

In the RBM, processing of sensory input consists of a linear-nonlinear transformation of a stimulus vector (*v*) that determines the probability that units in the hidden layer ‘spike’ (i.e., emit a ‘1’):(1)$$\begin{array}{cc} \mathrm{Pr}(h_{i}=1) & =\sigma (W_{i}v+B_{h_{i}}), \end{array}$$
where $\sigma \left(a\right)={\left[1+\exp\left(-a\right)\right]}^{-1}$ is a logistic sigmoid nonlinearity, *W* is a matrix of ‘synaptic’ weights between the visible and hidden layers, and $B_{h_{i}}$ is a per-unit bias that sets the baseline firing rate for hidden units $h_{i}$.

In addition to retaining phenomenological aspects of spiking population coding, RBMs can be trained readily using the contrastive divergence algorithm [26,27]. We trained RBMs on binarized regions of natural images in order to study emergent learned encoding strategies (Figure 1a; Methods Section 4.1 and Section 4.2). We evaluated a range of population sizes for the hidden layer (Figure 1b–e) to study how encoding strategies change with network size. Small networks did not accurately model the stimulus distribution (Figure 1b,c), and a minimum population size was necessary to faithfully model small binary image patches from the CIFAR dataset. Network activity became increasingly sparse (Figure 1d) and uncorrelated (Figure 1e) for larger hidden-unit population sizes, mimicking the sparse spiking activity of biological neuronal networks.

It appears that sufficiently large RBMs can learn stochastic spiking representations of incoming stimuli. We next examine these learned encoding strategies to answer two related questions. First, can we understand general principles of sensory encoding based on the strategies learned by these models? Second, are there statistical correlates of a model being ‘sufficiently large’ that could be used to identify the minimum population size required for good representations?

### 2.2. RBMs Provide an Energy-Based Interpretation of Spiking Population Codes

For models that capture the stimulus distribution well, we would like to understand how the network allocates its coding space: how do ‘visible’ stimuli map to spiking patterns in the latent ‘hidden’ layer, and vice-versa? The limited number of hidden units favors precise neural codes, in which specific stimuli reliably evoke a specific pattern of neuronal spiking. However, noise can limit coding precision, requiring multiple neural states to map to each stimulus to achieve robustness [9].

Overall, two strategies are available for increasing information content in stochastic spiking codes. Neurons can become reliable, and use precise codes with less noise. Neurons can also increase their firing rates. These strategies have natural analogues in information theory. Increasing codeword precision amounts to decreasing the conditional entropy of evoked neural activity, i.e., reducing the channel noise. Using higher firing rates amounts to increasing the ‘energy’ of the neural codes, which is equivalent to using longer symbols (or more bandwidth) in a conventional digital code. Hinton et al. (1995) [28] first noted this in the context of spiking latent-variable models, showing that in optimal codes the amount of information in a stimulus should match the average information in the latent spiking pattern minus the entropy (i.e., variability) of the evoked patterns. However, the question remains of how an optimized spiking channel might make use of these two encoding strategies.

Here, we assume that the sensory channel represents all stimuli equally, so that the amount of behaviorally-relevant information in each stimulus is indeed equal to its negative log-probability. This reflects the number of bits required to communicate it in an optimal code in Shannon sense. In reality, early stages of sensory processing filter and discard information, preserving only important details. This issue is minor, however, since one can consider the stimuli in the simulations here as reflecting only the behaviorally-relevant bits.

To explore this, let us first make precise these notions of ‘energy’ and ‘entropy’ in the trained RBM networks. For an RBM with weight matrix *W* and hidden and visible biases $B_{h}$ and $B_{v}$, the probability of any population activity state (h,v) can be written as:(2)$$\begin{array}{cc} \mathrm{Pr}(h,v) & =\exp\left(-E(h,v)\right)\\ E(h,v) & =-B_{v}^{⊤}v-B_{h}^{⊤}h-h^{⊤}Wv+\mathrm{constant}, \end{array}$$
where E(h,v) is the energy of the state (h,v). Throughout this paper, we will use the term ‘energy‘ synonymously with negative log-probability.

We adopt the compact notation of Dayan et al. (1995) [29], and write energy of a state (h,v) as $E_{h,v}^{\phi }$, where $\phi =\{W,B_{h},B_{v}\}$ are the model parameters. Probabilities are denoted similarly, and we use Q to denote the distribution of latent factors learned by the RBM network. In this notation, the stimulus-evoked entropy $H_{h|v}^{\phi }$ of the hidden-unit spiking *h* given a specific stimulus *v* is
(3)$$H_{h|v}^{\phi }=\sum_{h}Q_{h|v}^{\phi }E_{h|v}^{\phi }={\langle E_{h|v}^{\phi }\rangle }_{h|v}$$

Above, $Q_{h|v}^{\phi }$ denotes the distribution of activity patterns in the latent units evoked by stimulus *v*, and ${\langle \cdot \rangle }_{h|v}$ denotes expectation with respect to this distribution. $E_{h|v}^{\phi }$ denotes the ‘energy’ (negative log-probability) of a hidden pattern *h* given stimulus *v*. In this notation, optimal representations are achieved when the amount of information in a stimulus ($E_{v}$) matches the amount of information in the evoked spiking activity minus the entropy of any noise in the channel ($H_{h|v}$): (4)$$E_{v}={\langle E_{h,v}\rangle }_{h|v}-H_{h|v}.$$
(c.f. Equation (Equation 5) in Hinton et al. 1995 [28]). In practice, the optimization procedure identifies parameters ϕ that only approximately achieve the above relationship. For models that are too small, not all stimuli are equally well-encoded, as reflected in the increased Kullback-Leibler divergence in the fits for smaller models (Figure 1b).

### 2.3. Stimulus-Dependent Variability Suppression Is a Key Feature of Optimal Encoding

We calculated the stimulus-evoked energy and entropy for a range of network sizes (Methods Section 4.3). In Figure 2 we examine how these quantities vary a function of stimulus energy. Here, stimulus energy is equivalent to negative log-probability ($E_{v}=-\log P_{v}$), and reflects the amount of information (in bits) needed to specify a particular stimulus. Groups of stimuli with similar energy therefore reflect different bitrates required of the sensory communication channel.

We found that RBMs learned to reserve the highest-bandwidth (low noise) parts of coding space for high-information stimuli. This can be seen in Figure 2a, which shows that the stimulus-evoked entropy in the latent spiking activity is reduced when higher bitrates are needed, provided the channel is sufficiently large. This reflects an adaptive code that lowers neuronal variability when more bandwidth is required. Conversely, stimuli that require less bandwidth are represented using noisier parts of encoding space.

To communicate more information, neural codes can either reduce noise ($H_{h|v}$), or they can use more informative code-words. In optimal Shannon coding, more informative codewords are simply rarer (information is negative log-probability), and correspond to specific spiking patterns reserved for rare stimuli. One can summarize how ‘rare’ the spiking patterns for a particular stimulus are in terms of the average energy of the evoked codewords, which we denote as ${\langle E_{h}^{\phi }\rangle }_{h|v}$. Intuitively, ${\langle E_{h}^{\phi }\rangle }_{h|v}$ is the average number of bits needed to specify a particular codeword *h* evoked by stimulus *v* if we do not know *v* in advance.

We expected ${\langle E_{h}^{\phi }\rangle }_{h|v}$ to increase for higher stimulus bitrates, but found instead that it closely tracked variability ($H_{h|v}$), decreasing for stimuli that required more bits to communicate. Indeed, above a certain model size (∼35 units in this case), the stimulus-evoked entropy and energy tracked each-other with a 1:1 ratio. This is illustrated in Figure 2b, which plots the difference between these two quantities over a range of stimulus bitrates. Surprisingly, this 1:1 balance between energy and entropy corresponds to statistical criticality and the emergence of 1/f power-law statistics in the latent spiking activity. Criticality in the brain has been the subject of some controversy over the past decades [30,31], and we unpack this observation in more depth in the following sections.

### 2.4. Optimal Codes Exhibit Statistical Criticality

When we say that a collection of observations exhibit statistical criticality, we mean that they are consistent with being generated by a physical process that lies close to a phase transition in the thermodynamic sense. At first glance, it is unclear how the allocation of codewords in a stochastic spiking code relates to criticality, or why this relationship might be interesting from the standpoint of neural coding.

Historically, the study of statistical criticality in neural systems was motivated by theories that suggest that dynamical regimes close to a phase transition might be useful for processing information [14]. Indeed, several studies have suggested evidence of statistical criticality in neural data [32,33,34]. However, other studies call the significance of this into question [35], showing that these statistics can arise under very generic circumstances [24], might be inherited from the environment [36], and could even be a data-processing artefact [37] or due to insufficient sample sizes [38]. We hope to clarify some of this controversy by examining the emergence of statistical criticality in this in silico model of spiking population coding.

Figure 3 illustrates how the energy and entropy of stimulus-evoked activity varies as a function of stimulus bitrate. We group stimuli into sets $V_{E}$ of similar energy, which correspond to different bitrates required of the spiking channel. For the stimulus ensembles explored here, $E_{v}$ ranged from 6 to 20 bits per stimulus. In Figure 3 and Figure 4 we divide this range of energies evenly into bins, grouping stimuli with similar $E_{v}$ into a set $V_{E}$ for each bin. For each $V_{E}$, we plot the average stimulus-evoked entropy $H_{h|v}$ (a correlate of the spiking noise), and energy ${\langle E_{h}^{\phi }\rangle }_{h|v}$ (the average number of bits required to specify a particular evoked spiking pattern). To more clearly illustrate the scaling, the entropy is shifted by a constant $I_{\mathrm{enc}}$, which reflects the average difference between energy and entropy. For this particular set of stimuli, models with at least 35 hidden units exhibited a positive correlation between energy and entropy, with a slope that approaches one as the model size increases.

This relationship corresponds to the so-called “Zipf’s law” [31]. Zipf’s law refers to the frequency (*f*) of symbols in a dataset. Here, the symbols are the spiking codewords (*h*) in the hidden units. Zipf’s law states that frequency of a symbol is inversely proportional to its rank in the frequency table, i.e., the rarer a pattern is the more patterns there are of similar frequency. For example, in a dataset exhibiting Zipf’s law we would expect approximately $2^{E}$ patterns with frequency above $2^{-E}$ (up to some multiplicative constant). These statistics are especially curious in the context of the RBM, which can be interpreted as a type of Ising spin model. Ising spin models at a critical point exhibit Zipf’s law in their distribution of states [14,31].

We confirm that the 1:1 variation in entropy and energy observed here corresponds to Zipf’s law in the codeword frequencies in Figure 5a. The stimulus-evoked entropy $H_{h|v}$ determines the number of hidden codewords *h* that correspond to a given stimulus *v*. Loosely, one can think of a stimulus as eliciting $2^{H_{h|v}}$ possible patterns. Likewise, the energy of a hidden codeword $E_{h}$ is proportional to its negative log-probability.

In the models examined here, the energy and entropy of stimulus-evoked patterns vary similarly as a function of stimulus energy $E_{v}$, giving rise to Zipf’s law in the frequencies of population spiking patterns. This means that, as the neural code gets noisier, the probability of any specific codeword also decreases. Overall then, we find that stimuli encoded in the noisier parts of coding space are allocated over a larger pool of increasingly rare, but representationally equivalent, codewords. This strategy is essential for reserving the reliable parts of the coding space for high-information stimuli, while also using a robust code to communicate low-information stimuli.

Here we show that statistical criticality emerges naturally in a model of stochastic spiking encoding, but only for models that are large enough to capture the stimulus distribution. Our use of a ground-truth model simulation ensures that these statistics are not an artefact of recording or data-processing [37,38]. A natural question, however, is whether these statistics arise from the statistics of natural images, which also exhibit Zipf’s law [40]. In Figure 4 we confirm that this is not the case, as models trained on stimuli designed to have other statistics still exhibit 1/f power-law statistics in latent unit activity. Many processes can generate similar statistics, and while criticality implies 1/f statistics, the converse is not necessarily true [23,24,41]. We next therefore asked whether the observed statistics are associated with true criticality in the thermodynamic sense, and whether this tells us anything significant about the model optimization and the learned encoding strategies.

### 2.5. Statistical Correlates of the Size-Accuracy Trade-Off

So far, we have demonstrated that Zipf’s law emerges in optimized RBM models of spiking population codes. Do these statistics imply anything meaningful about the underling spiking population code, or could they arise from more mundane explanations [24,30]? To address these questions in depth, we leverage the fact that the RBM can be interpreted as a thermodynamic system. This means that one can define signatures of a true phase transition, and therefore examine whether these critical statistics imply anything meaningful with respect to model parameters and their optimization.

To explore the thermodynamic interpretation of the RBM, one can extend the energy-based definition of the RBM (Equation (Equation 2)) to include an inverse temperature parameter β=1/T:(5)$$P_{h,v}\propto \exp\left(-\beta E_{h,v}\right),$$
where $P_{h,v}$ is the probability of simultaneously observing stimulus *v* and hidden-layer output *h*, and $E_{h,v}$ is the negative log-probability at the trained model parameters with temperature β=1. This corresponds to scaling the biases and weights by β, and controls a single direction in parameter space that determines how ordered or disordered the spiking activity is. High temperatures (β→0) corresponding to a noisy phase where the probability of all states are equal. Low temperatures (β→∞) exhibit only a few fixed patterns. Critical models exists at a specific critical temperature $T_{c}$ that defines a transition between the these two phases.

To generalize this idea, we can study the Fisher Information Matrix (FIM), which defines a local measure of importance to various directions in the space of RBM parameters $\phi =\{W,B_{h},B_{v}\}$. The FIM provides an infinitesimal equivalent of the Kullback-Leibler divergence between the model and a neighbouring model, which differs by an infinitesimal deviation in the parameter space. For given model parameters $\phi =\{\phi_{1},\ldots ,\phi_{n}\}$, the entry $F_{ij}\left(\phi \right)$ in the FIM for the *i*th and *j*th parameters is defined as:(6)$$F_{ij}\left(\phi \right)=\sum_{v,h}P_{v,h}\frac{\partial^{2}E_{v,h}}{\partial \phi_{i}\partial \phi_{j}}.$$

For RBMs, one can calculate the FIM from the activity statistics (Methods Section 4.4). The FIM is a generalized measure of susceptibility or specific heat [42], and it diverges at the point of phase transition $\beta =1/T_{c}$. Intuitively, this is because the model’s statistics change abruptly at the critical temperature, where the model’s behavior as a function of parameters approaches a non-differentiable point with infinite curvature (i.e., diverging FIM) for increasing system sizes. For small, finite models, there is no true phase transition per-se. Instead, the FIM exhibits a local peak around $\beta =1/T_{c}$ which indicates the finite-size analogue of a critical temperature [42].

One can assess whether a given model is close to a phase transition by examining the structure of the FIM for a range of temperatures. Analyzing the behavior of the largest FIM eigenvalue is analogous to studying the divergence of specific heat [14], but its interpretation is more general. In Figure 5a we find that a local peak in the maximum FIM eigenvalue (the direction in parameter space with the largest curvature) emerges for models with >30 units. This is also the model size at which statistical criticality emerges (Figure 3 and Figure 5a rightmost column). We conclude that the emergence of statistical criticality corresponds to a true critical point in the thermodynamics sense. Empirically we find that models that are sufficiently large to fit the data exhibit a localized peak in the FIM curvature for β=1. We conjecture that these statistics might be useful in identifying the optimal model size that balances accuracy vs. size cost.

Above the model size at which criticality emerges (‘critical model size’), we find diminishing returns in terms of model accuracy (Figure 1b,c). We examined the structure of the FIM to determine whether the model exhibited ‘sloppy’ [43,44,45] parameters that might be removed without degrading accuracy. Indeed, we found that many single units or weights become relatively unimportant in larger models (Figure 5a). This suggests that the activity statistics may reveal superfluous neurons or synapses that could be removed or ‘pruned’ with relatively little damage to the network’s function. However, the parameter importance as assessed by FIM should be interpreted with caution. We find that the least ‘important’ units, in terms of FIM curvature, have receptive fields corresponding to complex or high spatial-frequency features (Figure 5d). These units therefore encode fine details of images. While removing a single unit might have a minor effect of the model accuracy, collectively many unimportant units may be necessary for maximising the encoded information.

We conclude, therefore, that the emergence of Zipf’s law in these simulations can be connected to an energy-based description of spiking correlations that lies close to a phase transition, and is not inherited from the statistics of the stimuli, nor is it a data-processing artefact. Furthermore, these statistics emerge at or around the point where adding additional neurons to the model leads to only marginal improvements in representational accuracy.

## 3. Discussion

Understanding neural population codes in the context of communications theory is challenging, since stochastic spiking channels differ in many aspects from the communications channels studied in engineering. In this work, we used restricted Boltzmann machines to study optimal encoding in stochastic spiking channels. Analogously to sensory systems, such models learn to encode the latent causes of incoming stimuli in terms of a stochastic binary representation. Although different stimuli require different number of bits to encode, the number of hidden units available for this representation is fixed, and different parts of the encoding space exhibit more channel noise than others.

Under these constraints, RBMs learned to represent higher bitrate stimuli by suppressing variability, which mirrors the behavior of in vivo neural populations [6]. Surprisingly, we found that high-information stimuli were often associated with lower energy code-words, a result which may connect to the synergy-by-silence observed in the retina [46]. This coding strategy can be explained by a competitive allocation of encoding space in a stochastic channel. Noise is largest when neurons are close to firing threshold, and so the noisiest parts of activity space exhibit intermediate firing rates. To handle higher bitrates it is necessary to signal reliably, and therefore avoid overlapping with these noisy parts of the coding space. Suppressing firing in a selective population of cells is one way to achieve this.

A central prediction of this coding strategy is that common (low information) stimuli are associated with less precise (more noisy) encoding. It would be interesting to revisit data recorded from sensory systems such as the retina, to see if the effective stimulus bitrate predicts the observed neuronal variability. Indeed, emerging results appear to be consistent with this scenario [47,48]. This result also highlights that that the fundamental unit of ‘neural coding’ is not a specific pattern of spiking activity *per se*. We found that many stimuli can be encoded by a large volume of equivalent spiking population codewords. The equivalence between different evoked spiking patterns ensures robust representations despite noise. This motivates further study to examine the extent to which noise-robustness strategies learned in stochastic latent-variable models resemble those observed in retinal population codes in vivo.

Spiking systems can also exhibit statistical criticality in the sparse, large-network limit [14]. In contrast, the statistical criticality observed here emerges abruptly at a finite optimal model size, which depends on the data being encoded (Figure 4). In these models, the combination of variability suppression and statistical criticality may be connected to the mechanism of Aitchison et al. [24], which notes that that 1/f power-law statistics arise whenever the observed data are generated from a mixture of hidden underlying causes. Our modelling work reveals a specific example of this phenomenon in systems that encode the external world. Here, we found that the bitrate of the underlying stimulus is the underlying, unobserved variable.

Theoretical work predicts that critical 1/f statistics might be common in latent-variable models like the RBM that are fit to data [21,22,38]. An important distinction between our study and previous ones is that we are not using criticality in an RBM fit to data to test whether the data themselves were generated by a critical process. Indeed, we reproduce the results of Mastromatteo and Marsili [22] in showing that the trained RBMs lie close to $T_{c}$ even when trained on outputs from Ising spin models that are far from criticality. Instead, we conjecture that statistical criticality correlates with the emergence of efficient codes. If sensory encoding can also be interpreted as a stochastic latent-variable model, then we should expect 1/f statistics as a default behavior in sensory systems, similar to that observed in machine learning models. If statistical criticality is, in a sense, the default, this implies that there is something interesting about models that do not exhibit it. We found that the absence of criticality was a symptom of a model being too small to properly explain the stimulus distribution. More generally, departure from 1/f statistics may reveal important clues about physiological constraints on, or the operating regime of, stochastic spiking channels.

In conclusion, we found that statistical machine learning models of spiking communication employ variability-suppression as an optimal encoding strategy. This is a very general phenomenon that must occur if a noisy spiking channel is to communicate stimuli with variable bitrates. We also found that successful encoding of the stimulus statistics correlates with the emergence of 1/f statistics in the frequency of population spiking ‘codewords’. These statistical signatures may be useful in identifying optimal model sizes in machine learning, and may provide clues about the operating regime of biological neural networks. While neuronal networks in vivo do not use the parameter optimization procedure that we used here, any learning procedure that adapts its internal states to optimally encode the external world should (approximately) optimize representational cost (Equation (Equation 4); ‘free-energy’ minimization [28,29,49,50]). For example, an energetic cost on neuronal reliability is formally related to free-energy minimization [51]. Our findings might also relate to work that finds emergent critical statistics at an optimal layer depth in deep neural networks [52,53]. It remains to be seen whether the variability suppression and other statistics observed here corresponds to the structure of the neural code in vivo.

## 4. Materials and Methods

### 4.1. Datasets

Images from the CIFAR-10 [54] data set were converted to gray scale, and binarized around the median pixel intensity. 90,000 randomly-selected circular patches of different radii were used as training data (Figure 1a).

### 4.2. Restricted Boltzmann Machines

RBMs were fit using one-step contrastive divergence (CD1) [26,55] implemented in Theano (github.com/martinosorb/rbm_utils) on NVIDIA GeForce GTX 980 GPUs. The learning rate was reduced in stages: 0.2, 0.1, 0.05, 0.01, $5\cdot 10^{-3}$, $10^{-3}$. 8 epochs were trained at each rate with mini-batch size 4. To estimate model energies, 350,000 states were sampled via 500 chains of Gibbs sampling, keeping one sample every 150 steps.

### 4.3. Energy and Entropy

In the RBM, hidden-layer entropy conditioned on stimulus *v* can be calculated in closed form as:$$H_{h|v}=\sum_{i=1\ldots N_{h}}g\left(a_{h|v}^{i}\right)-a_{h|v}^{i}f\left(a_{h|v}^{i}\right),$$
where $N_{h}$ is the number of hidden units, $a_{h|v}\;=\;v^{⊤}W+B_{h}$ is the stimulus-conditioned hidden-layer activation vector, $f\left(x\right)\;=\;1/\left(1+e^{-x}\right)$ is the sigmoid function, and $g\left(x\right)\;=\;\log\left(1+e^{x}\right)$. The expected conditional energy ${\langle E_{h}\rangle }_{h|v}$ is computed via sampling, where each individual $E_{h}$ is computed, up to a constant, as:$$E_{h}=-B_{h}h-\sum_{i=1\ldots N_{v}}g\left(W^{i}h+B_{v}^{i}\right)+\mathrm{const}.,$$
where $N_{v}$ is the number of visible units, $B_{v}$ is the vector of visible biases and $W^{i}$ is the row of the weight matrix associated with the *i*th visible unit. Energies are normalized using the energy of the lowest-energy (most frequent) pattern, estimated by sampling.

### 4.4. Fisher Information

The Fisher information matrix (FIM, Equation (Equation 6)) is a positive semidefinite matrix that defines the curvature of a metric on the manifold of parameters, and indicates the sensitivity of the model to parameter changes. Divergence of an eigenvalue of the FIM indicates an abrupt change in the model distribution, i.e., a phase transition. The FIM generalizes susceptibility and specific heat, physical quantities that diverge at critical points. For a vector $\vec{w}$ in parameter space, we define sensitivity as
$$S\left(\vec{w}\right)=\sqrt{{\vec{w}}^{T}F\vec{w}}.$$

The distribution of parameter sensitivity has in itself attracted interest [43,44]. For directions corresponding to eigenvectors of the Fisher information, the sensitivity is the square root of the corresponding eigenvalue. For changes in the *k*th parameter, $S_{k}=\sqrt{F_{kk}}$. In the case of RBMs (Equation (Equation 2)), we can consider the definition of the FIM (Equation (Equation 6)) with the biases and weights being possible values of ϕ. Expanding the derivatives, one gets to FIM entries of the form
$$\begin{array}{cc} F_{w_{ij},w_{kl}} & =\langle v_{i}h_{j}v_{k}h_{l}\rangle -\langle v_{i}h_{j}\rangle \langle v_{k}h_{l}\rangle \\ F_{w_{ij},b_{k}^{v}} & =\langle v_{i}h_{j}v_{k}\rangle -\langle v_{i}h_{j}\rangle \langle v_{k}\rangle \\ F_{w_{ij},b_{k}^{h}} & =\langle v_{i}h_{j}h_{k}\rangle -\langle v_{i}h_{j}\rangle \langle h_{k}\rangle \end{array}\;\;\;\;\begin{array}{cc} F_{b_{i}^{v},b_{k}^{h}} & =\langle v_{i}h_{k}\rangle -\langle v_{i}\rangle \langle h_{k}\rangle \\ F_{b_{i}^{v},b_{k}^{v}} & =\langle v_{i}v_{k}\rangle -\langle v_{i}\rangle \langle v_{k}\rangle \\ F_{b_{i}^{h},b_{k}^{h}} & =\langle h_{i}h_{k}\rangle -\langle h_{i}\rangle \langle h_{k}\rangle , \end{array}$$
where the brackets indicate averaging over the distribution Pr(v,h); these can be computed by sampling. The FIM diagonal summarizes the importance of individual units, and can be computed from locally-available variances and covariances:$$F_{b_{i}^{v},b_{i}^{v}}=\sigma_{v_{i}}^{2},\;\;\;\;\;\;\;F_{b_{i}^{h},b_{i}^{h}}=\sigma_{h_{i}}^{2},\;\;\;\;\;\;\;F_{w_{ij},w_{ij}}=\langle v_{i}^{2}h_{j}^{2}\rangle -{\langle v_{i}h_{j}\rangle }^{2}.$$

### 4.5. Free Energy in RBMs

We review the derivation of free energy in the context of RBMs [28]. Consider the problem of approximating a data distribution $P_{v}$ with a model distribution $Q_{v}^{\phi }$ parameterized by ϕ. In a latent variable model, one identifies a distribution on latent factors $Q_{h}^{\phi }$, as well as a mapping from latent factors to data patterns $Q_{v|h}^{\phi }$. The latent variables approximate the distribution over the data, i.e.,
$$Q_{v}^{\phi }=\sum_{h}Q_{h,v}^{\phi }=\sum_{h}Q_{v|h}^{\phi }Q_{h}^{\phi }.$$

Such a model model can be optimized by minimizing the negative log-likelihood of data given model parameters:$$\underset{\theta }{\mathrm{argmin}}\left[-\sum_{v}P_{v}\log Q_{v}^{\phi }\right]=\underset{\theta }{\mathrm{argmin}}\left[-\sum_{v}P_{v}\log\sum_{h}Q_{h,v}^{\phi }\right].$$

Jensen’s inequality provides an upper bound on the negative log-likelihood that can be easier to minimize. This minimization is equivalent to minimizing the KL divergence from the model to the data distribution:$$\begin{array}{cc} -\sum_{v}P_{v}\log\sum_{h}Q_{h,v}^{\phi }\; & =-\sum_{v}P_{v}\log\sum_{h}Q_{h|v}^{\phi }\frac{Q_{h,v}^{\phi }}{Q_{h|v}^{\phi }}\\ \; & \leq \sum_{v}P_{v}\underset{E_{v}^{\phi }}{\underset{︸}{\left[-\sum_{h}Q_{h|v}^{\phi }\log\frac{Q_{h,v}^{\phi }}{Q_{h|v}^{\phi }}\right]}}. \end{array}$$

This connects to the free-energy equation derived by Hinton et al. [28], which highlights the relationship between conditional distributions $Q_{h|v}^{\phi }$ and the visible pattern energies $E_{v}=-\log P_{v}$. When free energy is minimized over the data distribution, the model energies $E_{v}^{\phi }$ approximate the data energies and:$$\begin{array}{cc} E_{v}^{\phi } & =-\sum_{h}Q_{h|v}^{\phi }\log\frac{Q_{h,v}^{\phi }}{Q_{h|v}^{\phi }}\\ \; & =\underset{{\langle E_{h,v}^{\phi }\rangle }_{h|v}}{\underset{︸}{-\sum_{h}Q_{h|v}^{\phi }\log Q_{h,v}^{\phi }}}+\underset{-H_{h|v}^{\phi }}{\underset{︸}{\sum_{h}Q_{h|v}^{\phi }\log Q_{h|v}^{\phi }}} \end{array}$$

This relation is derived by Hinton et al. [28], Equation (Equation 5), from the perspective of minimizing communication cost, and in analogy to the Helmholtz free-energy from thermodynamics. This brief derivation illustrates the free-energy relationship in the context of minimizing an upper-bound on the negative log-likelihood of a latent-variable model. 

## Figures and Tables

**Figure 1 entropy-22-00714-f001:**
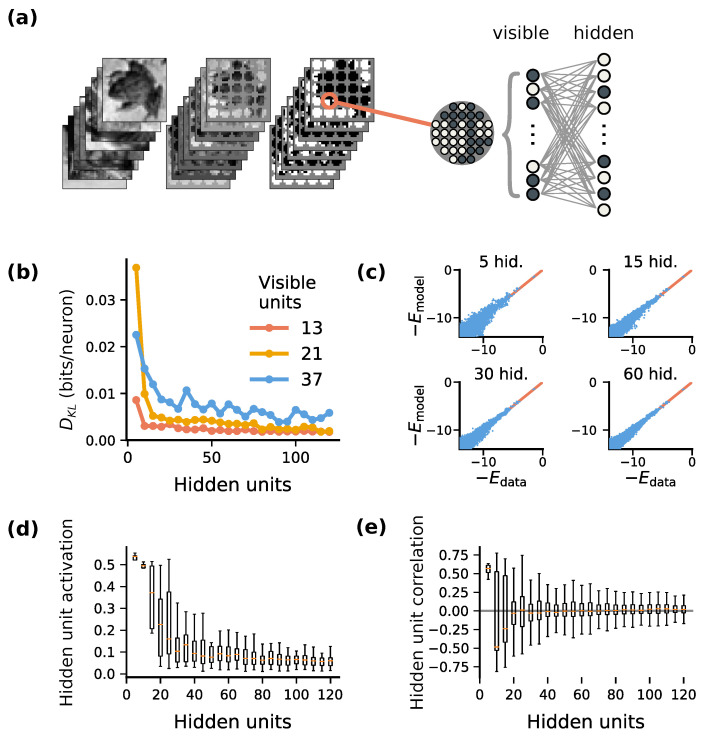
*Effect of the channel size on encoding of stimulus statistics:* (**a**) We trained RBMs to model local regions of (binarized) CIFAR-10 images. We interpret the number of hidden units as the size of a sensory communication channel. (**b**) A minimum number of hidden units is required to faithfully capture stimulus statistics. We quantified model accuracy by the Kullback-Leibler divergence between model samples and held-out training data. Accuracy improves as the hidden-layer size increases, up to a point. Results for three different sizes of stimulus patches (13, 21, 37 pixels) are shown. (**c**) Comparison of actual and predicted pattern probabilities for four hidden-layer sizes. We denote probability in terms of the negative log-probability (in bits), abbreviated as energy $E=-\log_{2}\mathrm{Pr}\left(\cdot \right)$. Larger models capture the stimulus distribution better. (**d**,**e**) Hidden-layer activation becomes sparser (**d**) as model size increases, and more decorrelated (**e**). 13 visible units were used for (**c**–**e**).

**Figure 2 entropy-22-00714-f002:**
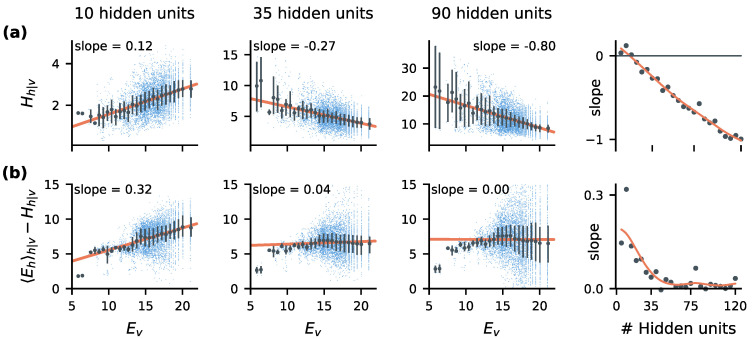
*Informative stimuli suppress variability in stochastic spiking communication channels:* We trained RBMs to encode 13 visible units from circular patches of binarized CIFAR-10 images. Left three plots show how the statistics of the evoked activity in hidden units (vertical axes) varies as a function of stimulus information content (horizontal axes), for various model sizes. Larger ‘energies’ ($E_{v}$) represent stimuli (blue dots) that require more bits to communicate. All units are in bits. Rightmost plots show the slope of the corresponding quantity in terms of $E_{v}$ as a function of model size. (**a**) Sufficiently large models learn to reduce channel entropy (variability) for stimuli that require more information to encode. (**b**) To communicate more information, neural codes can either reduce stimulus-conditioned entropy $H_{h|v}$, or they can use rarer code-words, i.e., increase ${\langle E_{h}\rangle }_{h|v}$. In sufficiently large models, we find that energy and entropy both decrease for stimuli that require more information to communicate. (gray bars; dots = mean, bars = inter-quartile range).

**Figure 3 entropy-22-00714-f003:**
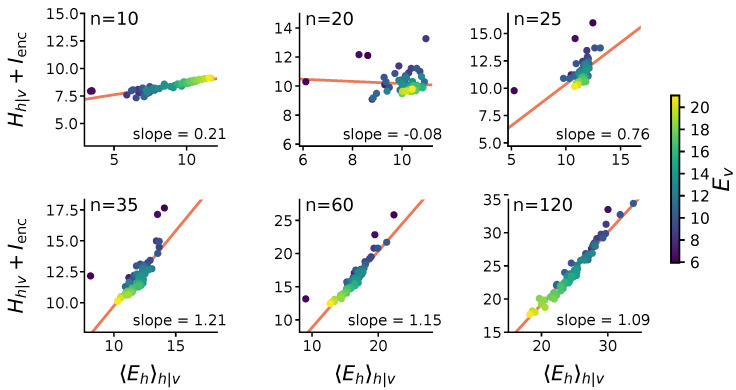
*Stimulus information content predicts energy and entropy of evoked activity in latent units:* Each plot shows the average stimulus-evoked entropy ($H_{h|v}$) plus a constant ($I_{\mathrm{enc}}$) on the vertical axis, against the information content of the code-words evoked by a given stimulus (${\langle E_{h}\rangle }_{h|v}$ horizontal axis). Here, $I_{\mathrm{enc}}=\langle D_{\mathrm{KL}}\left(Q_{h|v}∥Q_{h}\right)\rangle$ is the average energy-entropy relationship for all stimuli, which becomes approximately constant above a critical model size (Figure 2b). Color indicates the stimulus bitrate $E_{v}$. Points reflect the average energy and entropy of hidden patterns evoked for a given $E_{v}$. In too-small models (n = 10), low-variability states are used to represent common (low-information) stimuli. This relationship shifts as the encoding capacity increases (n = 20, 25). Above a critical model size (n ≥ 35), an inverse relationship between visible energies and the entropy of latent representations emerges: high-energy visible patterns suppress variability. A 1:1 trade-off between using energy and entropy for modulating bit rate also emerges (red lines). This relationship persists in larger models (n = 60, 120). This 1:1 trade-off reflects emergence of a 1/f power-law in the statistics of hidden unit activity, which gives rise to statistical criticality. Here, models were trained to encode 13 visible units from circular patches of binarized CIFAR-10 images.

**Figure 4 entropy-22-00714-f004:**
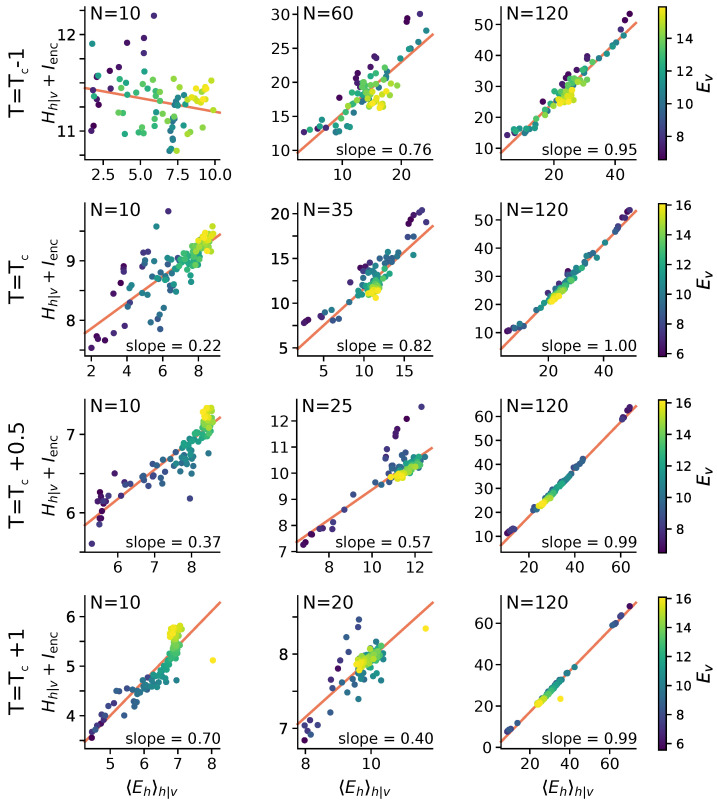
*Learned encoding strategies do not depend on the statistics of the stimulus distribution:* In natural visual stimuli, the visible samples themselves display 1/f power-law statistics. This might encourage similar statistics in the activations of hidden units, explaining the 1:1 trade-off between modulating entropy and energy that we observed. Here we show the energy-entropy balance as a function of stimulus information content (i.e., bit-rate, $E_{v}$) for RBMs fit to two-dimensional lattice Ising models, sampled at a range of temperature above and below the critical temperature of $T_{c}\;=\;2/\ln\left(1+\sqrt{2}\right)\;\approx \;$2.269. The energy-entropy balance converges to identity regardless of the data temperature (right column). However, the critical hidden-layer size (*N*) does decrease with temperature, illustrated here (middle column) by the increasing hidden-layer size displaying intermediate energy-entropy statistics. Small models (left column) exhibit a correlation between visible energy and entropy for training-data temperatures above $T_{c}$. Ising models were simulated on a 10 × 10 grid, and sampled via the Swendsen-Wang algorithm [39] with 10 k steps burn-in and 100 k training patterns drawn every 100 samples. 13-unit patches were presented to the RBM for training. All units are in bits.

**Figure 5 entropy-22-00714-f005:**
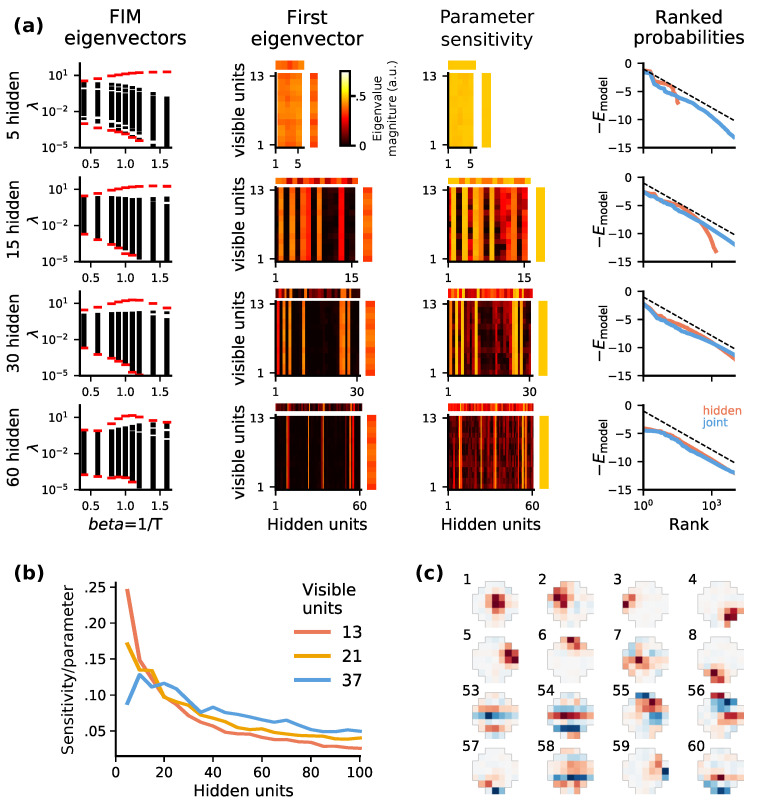
*Analyses of parameter sensitivity suggests an optimal model size for encoding sensory statistics:* (**a**) Analysis of the Fisher Information Matrix (FIM) over a range of hidden-layer sizes (top to bottom; 13 visible units). From left to right, (1) FIM eigenvalue spectra $\lambda_{i}$ (y-axis) over a range of inverse temperatures β indicate that model fits (β = 1) past a certain size lie at a peak in their generalized susceptibility. This is a correlate of criticality in Ising spin models. Eigenvalues below $10^{-5}$ are truncated, and the largest and smallest eigenvalues are in red; (2) Important parameters in the leading FIM eigenvector align with individual hidden units, and become sparse for larger hidden layers. The eigenvector is displayed separately for the weights (matrix), and the visible (vertical) and hidden (horizontal) biases; (3) The average sensitivity of each parameter over all FIM eigenvectors, shown here as the square root of the FIM diagonal, also shows sparsity, indicating that beyond a certain size additional hidden units contribute little to model accuracy. Data is shown as in column 2; (4) Variance of the hidden unit activation as a function of stimulus energy. In larger models, units with sensitive parameters contribute to encoding low energy, less informative patterns. (**b**) The average sensitivity of each parameter, measured by the trace of the FIM, normalized by hidden-layer size, decreases as hidden-layer size grows. (**c**) Hidden unit projective fields from a model with 37 visible and 60 hidden units, ordered by relative sensitivity (rank indicated above each image). More important units (ranks 1–8) encode spatially simple features such as localized patches, while the least important ones (ranks 53–60) have complex features.

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
