# Peer review of "Optimal Encoding in Stochastic Latent-Variable Models"

_entropy, 2020, doi:10.3390/e22070714_

Round 1
Reviewer 1 Report
The authors train Restricted Boltzmann Machines on binarised natural images and study the properties of the distribution of activity in the hidden layer, e.g. the conditional entropy of the hidden layer and its dependence on the energy of the observed etc, changing the number of hidden nodes.
In general I find the paper interesting. However, I have some concerns regarding the way this analysis is framed and interpreted, and some minor comments:
- I do not fully understand the emphases the authors make on relating this work to spikes and neural activity. The only similarity I can see is that spikes and the states of the observed nodes in an RBM are binary. But the differences between encoding sensory input in neural spiking patterns and the RBM are so much more that I don't think such a huge emphasis is warranted. I think the authors should present their results primarily as a study of the coding properties of an RBM, and then perhaps in the discussion discuss the possible relations to neural coding.
2. The relation to criticality also should be discussed with more caution. The authors show that the (generalised-)susceptibility diverges at beta=1 when they change beta. This is a dangerous procedure and has been shown in ref. 34 to indicate criticality when there is none. I think the authors presentation of Ref. 34 is not really accurate. They show that inferred models fit to data when going through the procedure of changing beta as the authors do here may show diverging susceptibility when there is none. The specific example they show is the Hawks process, in which the events are not binary or spin-like, but Poisson. The reason also is clearly discussed in that paper and it I certainly not by chance.
3. The authors mention studies that aim at relating the observation of heavy tailed distributions in some neural datasets to the statistics of the input. The fact that the 1/f distribution emerges for large enough number of hidden nodes could well be another piece of evidence supporting this idea. For large enough hidden layers, the hidden layer starts approaching the distribution of the observed nodes. Consider the limit of hidden layer being of a similar size as the observed layer. Then the hidden layer simply inherits the statistics of the observed to some degree. Perhaps it is already enough for the hidden layer in this data set to be of the size of 35 or so to capture the 1/f properties.
minor comments:
- starting from Eq. 3, the authors do not actually define Q_h|v, E_h|v and H_h|v. This lack of defending symbols and notations recurs throughout the paper. I think the authors should carefully read the paper and make sure that the notation is clarified.
- In Fig. 2, it would be nice to plot the slope as a function of the number of hidden nodes to see how smooth it goes from positive to negative.
- line 348: how do you exactly group? do you consider energy bands, if so how the size of the band chosen, or states in a group have exactly the same energy?
- I find the citation to ref 21 for neural activity inheriting its "critical" features from stimuli strange. This paper talks about how "critical features" appear in natural images. I think better citations are Tyrcha et al 2013 J Stat Mech which specifically explains Zipf's law in retinal data using input to the units and Ref. 31.
Author Response
Dear Reviewers,
We found your comments thought-provoking and constructive, and we are grateful for your kind, thoughtful, and rigorous appraisal of the manuscript.
Our responses to the comments of both reviewers follow. Please also see the attached document, which illustrates the changes to the manuscript in detail.
If we understood correctly, both reviewers shared concerns that (1) RBMs might not be an especially meaningful model in neuroscience, and (2) the connection to statistical criticality must be treated cautiously. We share these concerns. Yet we also argue that RBMs capture key aspects of neural sensory encoding, and can therefore provide a useful test bed to derive testable predictions. In particular, variability suppression in biological neural codes is an active area of research, and this work may constitute a useful theoretical complement. Regarding (2), our aim was indeed to treat statistical criticality with caution as an empirical result that correlates with the trade-off between network size and accuracy. To this end, we have made several edits to qualify these better, and added the following two paragraphs to the introduction, which address the plausibility of RBMs in theoretical neuroscience, and the interpretation of statistical criticality, respectively:
In this work, we used Restricted Boltzmann Machines (RBMs) to study encoding in stochastic spiking channels. RBMs consist of two layers of interconnected binary units, and the activity of these units is likened to the presence (in the case of a '1') or absence (in the case of a '0') of a spiking event in a single neuron in a specified brief time window. This simplified view of neurons as stochastic variables with discrete states also underlies ubiquitous mean-field models of neural population dynamics [@destexhe2009wilson].
RBMs balance biological realism and theoretical accessibility. The stochastic and binary nature of RBMs resembles physiological constraints on spiking communication, while the interpretation of RBMs as Ising spin models also allows access to information-theoretic and thermodynamic quantities [@schneidman2006weak; @shlens2006structure; @koster2014modeling; @tkavcik2015thermodynamics]. Although RBMs can be made to resemble spiking systems by extending their dynamics in time [@hinton2000spiking], statistical models of spiking populations commonly consider the synchronous case [@nasser2013spatio], which models only zero-lag dependencies between spikes. RBMs have been used as a statistical model to study retinal population coding [@zanotto2017modeling; @gardella2018blindfold], and multi-layer RBMs have been used as a model of retinal computation [@turcsany2014modelling]. RBMs can be connected to more biologically plausible spiking models [@shao2013linear], but added biological plausibility does not not change essential statistical features that we wish to study, and obscures relevant thermodynamic interpretations.
The analysis of these models from a thermodynamic viewpoint leads to the observation of scale-free ('$1/f$') statistics in the frequencies of the population codewords evoked by incoming stimuli. These scale-free statistics are often interpreted as a correlate of a phenomenon called 'statistical criticality', whose significance is a topic of debate in recent literature. It is important to qualify the sense in which statistical criticality is explored in this manuscript.
Critical statistics emerge naturally in large latent-variable models that are trained to represent external signals [@Schwab2014; @Mastromatteo2011]. They are not scientifically meaningful in isolation [@beggs2012being], and can arise from many other causes [@Aitchison2016; @touboul2017power]. In this work, the relevant connection is the *absence* of critical statistics in models that are too small to encode their inputs, and the emergence of these statistics above a certain model size.
REVIEWER 1
- I do not fully understand the emphasis the authors make on relating this work to spikes and neural activity. The only similarity I can see is that spikes and the states of the observed nodes in an RBM are binary. But the differences between encoding sensory input in neural spiking patterns and the RBM are so much more that I don't think such a huge emphasis is warranted. I think the authors should present their results primarily as a study of the coding properties of an RBM, and then perhaps in the discussion discuss the possible relations to neural coding.
Yes, to some extent we agree. However, theoretical neuroscience has a tradition of using abstract models to build qualitative intuition about neural population dynamics (since at least to the 1970s). The idea of using RBMs as a model for stochastic spiking communication was, to our understanding, present in Hinton and colleague’s original exposition of these models. The use of RBMs in modelling neural population coding is more recent, and made possible by advances in multi-electrode recordings. In computational neuroscience, RBMs have been used in two similar (but distinct) approaches. First, they have been used to examine the statistics of population codes, akin to a binary factor-analysis model. Second, they represent the simplest model of hierarchical processing in sensory systems (neglecting recurrent connections) fit under a maximum likelihood objective. It is this latter interpretation that we explore here.
It is said that a model should be as simple as possible, and no simpler. In this spirit, we selected the RBM because it was the minimal model that retained essential features of a neuronal sensory system, namely:
- Neurons have spatially localised (in stimulus space) receptive fields, and transmit information downstream using patterns of binary variables.
- Neuronal output can be considered over time-bins of finite width Δt, and stimuli can be viewed as being encoded in the binary patterns of population spiking.
- The number of output cells (in our work: N_h hidden units) is fixed
- Stimuli are communicated synchronously: every stimulus must be encoded within Δt amount of time, and must be communicated over N_h hidden units
- Outputs are binary (spikes) and stochastic
(these assumptions incorporated into a new paragraph in the main text, beginning line 77)
All of that being said, it is still possible that experiments will show that our attempt to use RBMs to say something generic about noisy sensory codes is misguided. We make a very specific, testable prediction: response variability should depend on the frequency of the stimulus pattern, with low variability expected for rare patterns. Informally this can be observed as, for instance, lower response variability in the retina when natural images are presented (MM Churchland et al. 2010, Nat Neurosci). We hope our work can inspire experimental testing.
To highlight this, we have added a new paragraph to the introduction (mentioned above).
- The relation to criticality also should be discussed with more caution. The authors show that the (generalised-)susceptibility diverges at beta=1 when they change beta. This is a dangerous procedure and has been shown in ref. 34 to indicate criticality when there is none. I think the authors presentation of Ref. 34 is not really accurate. They show that inferred models fit to data when going through the procedure of changing beta as the authors do here may show diverging susceptibility when there is none. The specific example they show is the Hawks process, in which the events are not binary or spin-like, but Poisson. The reason also is clearly discussed in that paper and it is certainly not by chance.
Yes, thank you. Our summary of (34) (Mastromatteo and Marsili) was wrong. We have corrected this throughout the paper to reflect that it is the inference procedure that encourages models to lie close to Tc. In addition to the new paragraph in the introduction (mentioned above), we have revised the discussion to read (starting line 297)
Theoretical work predicts that critical $1{/}f$ statistics might be common in latent-variable models like the RBM that are fit to data \citep{Mastromatteo2011, saremi2014criticality, Schwab2014}. An important distinction between our study and previous ones is that we are not using criticality in an RBM fit to data to test whether the data themselves were generated by a critical process. Indeed, we reproduce the results of Mastromatteo and Marsili \citep{Mastromatteo2011} in showing that the trained RBMs lie close to $T_c$ even when trained on outputs from Ising spin models that are far from criticality. Instead, we conjecture that statistical criticality correlates with the emergence of efficient codes. If sensory encoding can also be interpreted as a stochastic latent-variable model, then we should expect $1/f$ statistics as a default behavior in sensory systems, similar to that observed in machine learning models. If statistical criticality is, in a sense, the default, this implies that there is something interesting about models that \textit{do not} exhibit it. We found that the absence of criticality was a symptom of a model being too small to properly explain the stimulus distribution. More generally, departure from $1{/}f$ statistics may reveal important clues about physiological constraints on, or the operating regime of, stochastic spiking channels.
- The authors mention studies that aim at relating the observation of heavy tailed distributions in some neural datasets to the statistics of the input. The fact that the 1/f distribution emerges for large enough number of hidden nodes could well be another piece of evidence supporting this idea. For large enough hidden layers, the hidden layer starts approaching the distribution of the observed nodes. Consider the limit of hidden layer being of a similar size as the observed layer. Then the hidden layer simply inherits the statistics of the observed to some degree. Perhaps it is already enough for the hidden layer in this data set to be of the size of 35 or so to capture the 1/f properties.
We agree that, in principle, 1/f statistics in neural networks may be inherited from their inputs. The connection between model size and input size is curious, and we are not quite sure what to make of it. We believe our results to be more general. What we hope to highlight in this manuscript is the following:
- Statistical criticality is absent for models that are too small, even if the inputs are 1/f.
- Statistical criticality seems to emerge once models are large enough to explain the latent causes of their inputs, even if these inputs are not 1/f (Figure 4)
- The emergence of statistical criticality seems to be a result of model fitting (akin to Mastromatteo and Marsili’s results), and not the stimulus statistics.
Indeed, this paper began because we were using RBMs to study the statistics of retinal population codes. Dr. Sorbaro soon noticed that the fitted models were all close to Tc, even for control inputs that did not exhibit 1/f statistics. This led us to the Mastromatteo and Marsili result. We realized that the emergence of these statistics had less to do with the stimuli, and more to do with model optimization. We then also noticed the partitioning of the encoding space, which may be connected to variability suppression and other features of neural population coding.
Minor comments:
- starting from Eq. 3, the authors do not actually define Q_h|v, E_h|v and H_h|v. This lack of defending symbols and notations recurs throughout the paper. I think the authors should carefully read the paper and make sure that the notation is clarified.
Yes, thank you. We have updated the text surrounding equations 2, 5, and 6 to make variable definitions explicit.
- In Fig. 2, it would be nice to plot the slope as a function of the number of hidden nodes to see how smooth it goes from positive to negative.
We have added new subplots to Figure 2 to show how the slope changes as a function of network size.
- line 348: how do you exactly group? do you consider energy bands, if so how the size of the band chosen, or states in a group have exactly the same energy?
Yes, thank you. We have added clarification (line 170) in the text: For the stimulus ensembles explored here, E_v ranged from 6 to 20 bits per stimulus. In Figures 3 and 4 we divide this range of energies evenly into bins, grouping stimuli with similar (but non-identical) E_v into a set \mathcal{V}_E for each bin.
- I find the citation to ref 21 for neural activity inheriting its "critical" features from stimuli strange. This paper talks about how "critical features" appear in natural images. I think better citations are Tyrcha et al 2013 J Stat Mech which specifically explains Zipf's law in retinal data using input to the units and Ref. 31.
The rationale behind that citation was simply to refer to a proof that the visual environment is often found to have critical statistics. The citation suggested by the reviewer is indeed more appropriate, and we have used it instead. Thank you for the suggestion.
REVIEWER 2
This paper is thorough, but somehow, I don't walk away convinced that statistical criticality here is useful scientifically. This predisposition of mine emphatically does not affect my desire to recommend acceptance. I like the way that the authors said this in the Discussion, i.e. systems that do not exhibit criticality are more interesting, in that lack of criticality might be a signature of lack of adaptation. It might be useful to move that sentiment to the Introduction.
Yes, this is a good idea. To this end, we added the new paragraph in the introduction (mentioned above) qualifying the sense in which we discuss statistical criticality in this work.
Some might argue that RBMs are not thought to be great models of some undefined area of the brain. (Might be useful to define what area of the brain you think this might correspond to, if any.) So, I take this paper as more of a series of computer experiments on properties of responses of neural network architectures when subjected to various types of input.
Yes. We agree that the connection is quite abstract and perhaps worthy of scrutiny. However, it is (for better or for worse), relatively common for theoretical neuroscience to seek qualitative intuition from such abstract models. We hope that the newly-added introductory paragraph clarifies this (see above). See also our response to Reviewer 1: we feel that the RBM is, in some sense, the simplest model that is still suitable for exploring spiking population codes in the retina, as elaborated in (line 77 ff)
Minor comments:
- This is emphatically not an issue with the paper, but just is boggling my mind: what is going on with the small networks? The switchover from the behavior of small networks to large ones (Fig 2) is quite striking, but I can't think of an explanation-- is there a training error, in which finding the right parameters is easy with smaller networks? More on that…
To be honest, we do not fully understand the transition that occurs around the model size where 1/f statistics emerge. We know that too-small models fail to capture the stimulus statistics (as evidenced by the large Dkl from the training data, Figure 1b). If one refers to Hinton’s original “wake-sleep” paper, this condition correlates with the latent units failing to learn statistically independent latent explanations for the visible patterns. We know that too-small models are favoring the representation of common stimuli at the expense of rare ones: energies are matched better for more frequent patterns, and the number of bits lost (according to cross-entropy) per stimulus increases for rare stimuli. The emergence of 1/f statistics coincides with convergence of the model (in terms of Dkl), and further increasing model size does not lead to improved decoding (either in terms of Dkl or in terms of cross-entropy encoding loss). The best we can speculate is that, when there are insufficient hidden units, the training is “frustrated” in some sense. It is unable to factorize the inputs, and favors common stimuli at the expense of rare ones.
- Another maybe useful narrative for Fig 2: panel b shows that E_h - H_{h|v} doesn't correlate with E_v, at least in large enough networks. Alternatively, can think of this as E_h and H_{h|v} both having the same correlation with E_v. Since contrastive divergence tries to minimize E_h + E_v (maybe missing a term here), you'd expect a negative correlation between E_h and E_v. So then you have a positive correlation between E_v and H_{h|v}.
Yes, for sufficiently large models E_h and H_{h|v} both have the same correlation with E_v, and this corresponds to the emergence of statistical criticality. We must confess that our own attempts to explain this behavior, in terms of the energy-based description, were unsuccessful. We suspect that there is some underlying, simple, explanation. Our best qualitative guess is that efficient coding does not privilege either energy or entropy, and so if the model has sufficient capacity both of these quantities are used similarly to help match E_v. We were unable to derive a rigorous mathematical proof that this should be the case. We hope that other researchers who are more versed in theory might be able to paint a clearer picture.
- Missing a definition of E_{h|v}, since you only have a definition of E_{h,v}.
Yes. thank you; We have made definitions surrounding Eq. 3 more explicit (and Eqs. 5 and 6 as well)
- The subsection title "Evidence for an optimal population size" seems misleading, given that accuracy keeps on improving as you make the machine bigger, and the only nod to choosing a non-infinite machine is that there should be some undefined "cost size". That subsection is more of a further test of statistical criticality.
We have renamed this section “Statistical correlates of the size-accuracy trade-off”. Incidentally, we find that the model performance (in terms of encoded information) improves only marginally above the critical model size, which corresponds to the minimum number of hidden units needed to explain the latent structure in the visible layer.
- Maybe a useful link: https://pubmed.ncbi.nlm.nih.gov/24708368/
Thank you for the paper suggestion, we have added that reference to section 2.4.

Reviewer 2 Report
This paper uses Restricted Boltzmann Machines (RBMs) to model optimal neural codes, concluding that these RBMs operate at a statistical criticality and have particular encoding strategies reminiscent of strategies that have been found in real brains.
This paper is thorough, but somehow, I don't walk away convinced that statistical criticality here is useful scientifically. This predisposition of mine emphatically does not affect my desire to recommend acceptance. I like the way that the authors said this in the Discussion, i.e. systems that do not exhibit criticality are more interesting, in that lack of criticality might be a signature of lack of adaptation. It might be useful to move that sentiment to the Introduction.
Some might argue that RBMs are not thought to be great models of some undefined area of the brain. (Might be useful to define what area of the brain you think this might correspond to, if any.) So, I take this paper as more of a series of computer experiments on properties of responses of neural network architectures when subjected to various types of input.
Minor comments:
- This is emphatically not an issue with the paper, but just is boggling my mind: what is going on with the small networks? The switchover from the behavior of small networks to large ones (Fig 2) is quite striking, but I can't think of an explanation-- is there a training error, in which finding the right parameters is easy with smaller networks? More on that...
- Another maybe useful narrative for Fig 2: panel b shows that E_h - H_{h|v} doesn't correlate with E_v, at least in large enough networks. Alternatively, can think of this as E_h and H_{h|v} both having the same correlation with E_v. Since contrastive divergence tries to minimize E_h + E_v (maybe missing a term here), you'd expect a negative correlation between E_h and E_v. So then you have a positive correlation between E_v and H_{h|v}.
- Missing a definition of E_{h|v}, since you only have a definition of E_{h,v}.
- The subsection title "Evidence for an optimal population size" seems misleading, given that accuracy keeps on improving as you make the machine bigger, and the only nod to choosing a non-infinite machine is that there should be some undefined "cost size". That subsection is more of a further test of statistical criticality. Maybe a useful link: https://pubmed.ncbi.nlm.nih.gov/24708368/
Author Response

(The authors gave the same response as above.)
